# DuoRC: Towards Complex Language Understanding with Paraphrased Reading Comprehension

## Abstract

We propose DuoRC, a novel dataset for Reading Comprehension (RC) that motivates several new challenges for neural approaches in language understanding beyond those offered by existing RC datasets. DuoRC contains 186,089 unique question-answer pairs created from a collection of 7680 pairs of movie plots where each pair in the collection reflects two versions of the same movie - one from Wikipedia and the other from IMDb - written by two different authors. We asked crowdsourced workers to create questions from one version of the plot and a different set of workers to extract or synthesize answers from the other version. This unique characteristic of DuoRC where questions and answers are created from different versions of a document narrating the same underlying story, ensures by design, that there is very little lexical overlap between the questions created from one version and the segments containing the answer in the other version. Further, since the two versions have different levels of plot detail, narration style, vocabulary, *etc*., answering questions from the second version requires deeper language understanding and incorporating external background knowledge. Additionally, the narrative style of passages arising from movie plots (as opposed to typical descriptive passages in existing datasets) exhibits the need to perform complex reasoning over events across multiple sentences. Indeed, we observe that state-of-the-art neural RC models which have achieved near human performance on the SQuAD dataset (Rajpurkar et al., 2016b), even when coupled with traditional NLP techniques to address the challenges presented in DuoRC exhibit very poor performance (F1 score of 37.42% on DuoRC v/s 86% on SQuAD dataset). This opens up several interesting research avenues wherein DuoRC could complement other RC datasets to explore novel neural approaches for studying language understanding.

## 1 Introduction

Natural Language Understanding is widely accepted to be one of the key capabilities required for AI systems. Scientific progress on this endeavor is measured through multiple tasks such as machine translation, reading comprehension, question-answering, and others, each of which requires the machine to demonstrate the ability to "comprehend" the given textual input (apart from other aspects) and achieve their task-specific goals. In particular, Reading Comprehension (RC) systems are required to "understand" a given text passage as input and then answer questions based on it. *It is therefore critical, that the dataset benchmarks established for the RC task keep progressing in complexity to reflect the challenges that arise in true language understanding, thereby enabling the development of models and techniques to solve these challenges.*

For RC in particular, there has been significant progress over the recent years with several benchmark datasets, the most popular of which are the SQuAD dataset (Rajpurkar et al., 2016a), TriviaQA (Joshi et al., 2017), MS MARCO (Nguyen et al., 2016), MovieQA (Tapaswi et al., 2016) and cloze-style datasets(Mostafazadeh et al., 2016; Onishi et al., 2016; Hermann et al., 2015). However, these benchmarks, owing to both the nature of the passages and the question-answer pairs to evaluate the RC task, have 2 primary limitations in studying language understanding: (i) Other than MovieQA, which is a small dataset of 15K QA pairs, all other large-scale RC datasets deal only with factual descriptive passages and not narratives (involving events with causality linkages that require reasoning

**Movie: Twelve Monkeys**

**Original Plot Synopsis(Wikipedia)**

A deadly virus released in 1996...,[James Cole is a prisoner living in a subterranean compound beneath the ruins of Philadelphia.]$^{Q1}$ [Cole is selected for a mission]$^{Q2}$, ...

[Cole arrives in Baltimore]$^{Q3}$ in 1990, not 1996 as planned...[Goines denies any involvement with the group and says that in 1990 Cole originated the idea of wiping out humanity with a virus stolen from Goines' virologist father.]$^{Q4}$

Cole convinces himself... Railly confronts him with evidence of his time travel.. [They decide to spend their remaining time together in the Florida Keys before the onset of the plague]$^{Q5}$

[At the airport, Cole leaves a last message]$^{Q6}$ .... [He is soon confronted by Jose, an acquaintance from his own time, who gives Cole a handgun]$^{Q7}$ and ambiguously instructs him to follow orders. At the same time, Railly spots Dr. Peters....

Cole forces his way through a security checkpoint.... [Peters, aboard the plane with the virus]$^{Q8}$, ...

**Paraphrased Plot Synopsis (IMDB)**

A virus, deliberately released in 1996 … One such prisoner is [James Cole, who after retrieving samples is given the chance to go back in time to 1996]$^{Q2}$ and find information about the group believed responsible, known as "The Army of 12 Monkeys."

Throughout the ensuing episodes, Cole … There he meets Jeffrey Goines, … Cole is now racing against time… he wants to stay in 1996 with Dr. Railly, …They [travel to Philadelphia]$^{Q1}$,

[Jeffrey rambles about how Cole had given him the idea to release a virus that would destroy most of humanity.]$^{Q4}$ Cole leaves, ...and then posts flyers declaring "We did it!" [Cole realizes that the "Army" is not the threat, and he leaves a phone message to that effect]$^{Q6}$.

[Jose, a fellow "volunteer" from the present, approaches Cole with orders for him to complete his mission and hands him a revolver]$^{Q7}$... In an airport, while attempting with Cole to elude capture, Dr. Railly recognizes [Dr Peters, a man who worked with Jeffrey Goines's father …. The man goes through airport screening and manages to persuade security that his biological samples]$^{Q8}$...

**Q1:** James Cole is a prisoner living in a subterranean shelter beneath what city? Philadelphia, Philadelphia
**Q2:** What is the name of the person selected for the mission? James Cole, James Cole
**Q3:** Where did Cole arrive in 1990? Baltimore, -
**Q4:** Who does Goines claim came up with the idea to exterminate humanity? Cole, Cole
**Q5:** Where do Cole and Railly decide to go before the plague? Florida Keys, -
**Q6:** Where does Cole leave his message? At the airport, on the phone
**Q7:** Who gives Cole a handgun? Jose, Jose
**Q8:** Peters is aboard the plane with what? Virus, biological samples

**Figure 1:** Example QA pairs obtained from the original movie plot and the paraphrased plot. The relevant spans needed for answering the corresponding question are highlighted in blue and red with the respective question numbers. Note that the span highlighting shown here is for illustrative purposes only and is not available in the dataset.

and background knowledge) which is the case with a lot of real-world content such as story books, movies, news reports, etc. (ii) their questions possess a large lexical overlap with segments of the passage, or have a high noise level in Q/A pairs themselves. As demonstrated by recent work, this makes it easy for even simple keyword matching algorithms to achieve high accuracy (Weissenborn et al., 2017). In fact, these models have been shown to perform poorly in the presence of adversarially inserted sentences which have a high word overlap with the question but do not contain the answer (Jia & Liang, 2017). While this problem does not exist in TriviaQA it is admittedly noisy because of the use of distant supervision. Similarly, for cloze-style datasets, due to the automatic question generation process, it is very easy for current models to reach near human performance (Cui, 2017). This therefore limits the complexity in language understanding that a machine is required to demonstrate to do well on the RC task.

Motivated by these shortcomings and to push the state-of-the-art in language understanding in RC, in this paper we propose DuoRC, which specifically presents the following challenges beyond the existing datasets:

1. DuoRC is especially designed to contain a large number of questions with low lexical overlap between questions and their corresponding passages.

2. It requires the use of background and common-sense knowledge to arrive at the answer and go beyond the content of the passage itself.

3. It contains narrative passages from movie plots that require complex reasoning across multiple sentences to infer the answer.

4. Several of the questions in DuoRC, while seeming relevant, cannot actually be answered from the given passage, thereby requiring the machine to detect the *unanswerability* of questions.

In order to capture these four challenges, DuoRC contains QA pairs created from pairs of documents describing movie plots which were gathered as follows. Each document in a pair is a different version of the same movie plot written by different authors; one version of the plot is taken from the Wikipedia page of the movie whereas the other from its IMDb page (see Fig. 1 for portions of

an example pair of plots from the movie "Twelve Monkeys"). We first showed crowd workers on Amazon Mechanical Turk (AMT) the *first* version of the plot and asked them to create QA pairs from it. We then showed the *second* version of the plot along with the questions created from the *first* version to a different set of workers on AMT and asked them to provide answers by reading the second version only. Since the two versions contain different levels of plot detail, narration style, vocabulary, etc., answering questions from the second version exhibits all of the four challenges mentioned above.

We now make several interesting observations from the example in Fig. 1. For 4 out of the 8 questions (Q1, Q2, Q4, and Q7), though the answers extracted from the two plots are exactly the same, the analysis required to arrive at this answer is very different in the two cases. In particular, for Q1 even though there is no explicit mention of *the prisoner living in a subterranean shelter* and hence no lexical overlap with the question, the workers were still able to infer that the answer is *Philadelphia* because that is the city to which James Cole travels to for his mission. Another interesting characteristic of this dataset is that for a few questions (Q6, Q8) alternative but valid answers are obtained from the second plot. Further, note the kind of complex reasoning required for answering Q8 where the machine needs to resolve coreferences over multiple sentences (*that man* refers to *Dr. Peters*) and use common sense knowledge that if an item clears an airport screening, then a person can likely board the plane with it. To re-emphasize, these examples exhibit the need for machines to demonstrate new capabilities in RC such as: (i) employing a knowledge graph (e.g. to know that Philadelphia is a city in Q1), (ii) common-sense knowledge (e.g., *clearing airport security* implies *boarding*) (iii) paraphrase/semantic understanding (e.g. revolver is a type of handgun in Q7) (iv) multiple-sentence inferencing across events in the passage including coreference resolution of named entities and nouns, and (v) educated guesswork when the question is not directly answerable but there are subtle hints in the passage (as in Q1). Finally, for quite a few questions, there wasn't sufficient information in the second plot to obtain their answers. In such cases, the workers marked the question as "unanswerable". This brings out a very important challenge for machines to exhibit (i.e. detect *unanswerability* of questions) because a practical system should be able to know when it is not possible for it to answer a particular question given the data available to it, and in such cases, possibly delegate the task to a human instead.

Current RC systems built using existing datasets are far from possessing these capabilities to solve the above challenges. In Section 4, we seek to establish solid baselines for DuoRC employing state-of-the-art RC models coupled with a collection of standard NLP techniques to address few of the above challenges. Proposing novel neural models that solve all of the challenges in DuoRC is out of the scope of this paper. Our experiments demonstrate that when the existing state-of-the-art RC systems are trained and evaluated on DuoRC they perform poorly leaving a lot of scope for improvement and open new avenues for research in RC. Do note that this dataset is not a substitute for existing RC datasets but can be coupled with them to collectively address a large set of challenges in language understanding with RC (*the more the merrier*).

## 2 RELATED WORK

Over the past few years, there has been a surge in datasets for Reading Comprehension. Most of these datasets differ in the manner in which questions and answers are created. For example, in SQuAD (Rajpurkar et al., 2016a), NewsQA (Trischler et al., 2016), TriviaQA (Joshi et al., 2017) and MovieQA (Tapaswi et al., 2016) the answers correspond to a span in the document. MS-MARCO uses web queries as questions and the answers are synthesized by workers from documents relevant to the query. On the other hand, in most cloze-style datasets (Mostafazadeh et al., 2016; Onishi et al., 2016) the questions are created automatically by deleting a word/entity from a sentence. There are also some datasets for RC with multiple choice questions (Richardson et al., 2013; Berant et al., 2014; Lai et al., 2017) where the task is to select one among $k$ given candidate answers.

Given that there are already a few datasets for RC, a natural question to ask is "*Do we really need any more datasets?*". We believe that the answer to this question is *yes*. Each new dataset brings in new challenges and contributes towards building better QA systems. It keeps researchers on their toes and prevents research from stagnating once state-of-the-art results are achieved on one dataset. A classic example of this is the CoNLL NER dataset (Tjong Kim Sang & De Meulder, 2003). While several NER systems (Passos et al., 2014) gave close to human performance on this dataset, NER on general web text, domain specific text, noisy social media text is still an unsolved problem (mainly due to

the lack of representative datasets which cover the real-world challenges of NER). In this context, DuoRC presents 4 new challenges mentioned earlier which are not exhibited in existing RC datasets and would thus enable exploring novel neural approaches in complex language understanding. The hope is that all these datasets (including ours) will collectively help in addressing a wide range of challenges in QA and prevent stagnation via overfitting on a single dataset.

## 3 DATASET

In this section, we elaborate on our dataset collection process which consisted of the following three phrases.

1. **Extracting parallel movie plots:** We first collected top 40K movies from IMDb across different genres (crime, drama, comedy, etc.) whose plot synopsis were crawled from Wikipedia as well as IMDb. We retained only 7680 movies for which both the plots were available and longer than 100 words. In general, we found that the IMDb plots were usually longer (avg. length 926 words) and more descriptive than the Wikipedia plots (avg. length 580 words).

2. **Collecting QA pairs from shorter version of the plot (*SelfRC*):** As mentioned earlier, on average the longer version of the plot is almost double the size of the shorter version which is itself usually 500 words long. Intuitively, the longer version should have more details and the questions asked from the shorter version should be answerable from the longer one. Hence, we first showed the shorter version of the plot to workers on AMT and ask them to create QA pairs from it. For the answer, the workers were given freedom to either pick an answer which directly matches a span in the document or synthesize the answer from scratch. This option allowed them to be creative and ask hard questions where possible. We found that in 70% of the cases the workers picked an answer directly from the document and in 30% of the cases they synthesized the answer. We thus collected 85,773 such QA pairs along with their corresponding documents. We refer to this as the *SelfRC* dataset because the answers were derived from the same document from which the questions were asked.

3. **Collecting answers from longer version of the plot (*ParaphraseRC*):** We then paired the questions from the *SelfRC* dataset with the corresponding longer version of the plot and showed it to a different set of AMT workers asking them to answer these questions from the longer version of the plot. They now have the option of either (a) selecting an answer which matches a span in the longer version, or (b) synthesizing the answer from scratch, or (c) marking the question not-answerable because of lack of information in the given passage. We found that in 50% of the cases the workers selected an answer which matched a span in the document, whereas in 37% cases they synthesized the answer and in 13% cases they said that question was not answerable. The workers were strictly instructed to derive the answer from the plot and not rely on their personal knowledge about the movie (in any case given the large number of movies in our dataset the chance of a worker remembering all the plot details for a given movie is very less). Further, a wait period of 2-3 weeks was deliberately introduced between the two phases of data collection to ensure the availability of a fresh pool of workers as well as to reduce information bias among any worker common to both the tasks. We refer to this dataset, where the questions are taken from one version of the document and the answers are obtained from a different version, as *ParaphraseRC* dataset. We collected 100,316 such {*question, answer, document*} triplets.

Note that the number of unique questions in the *ParaphraseRC* dataset is the same as that in *SelfRC* because we do not create any new questions from the longer version of the plot. We end up with a greater number of {*question, answer, document*} triplets in *ParaphraseRC* as compared to *SelfRC* (100,316 v/s 85,773) since movies that are remakes of a previous movie had very little difference in their Wikipedia plots. Therefore, we did not separately collect questions from the Wikipedia plot of the remake. However, the IMDb plots of the two movies are very different and so we have two different longer versions of the movie (one for the original and one for the remake). We can thus pair the questions created from the Wikipedia plot with both the IMDb versions of the plot and hence we end up with more {*question, answer, document*} triplets.

| Question/Answer-Length Statistics | |
|---|---|
| Avg. Question-Length | 9 words |
| Avg. Answer Length (*SelfRC* Dataset) | 3 words |
| Avg. Answer Length (*ParaphraseRC* Dataset) | 5 words |
| Avg. Number of noun phrases in Question | 4 |
| Avg. Number of noun phrases in Answers | 1.5 |
| **Comparative Hardness of *ParaphraseRC* w.r.t *SelfRC*** | |
| % of QAs with no noun phrase common between Question and Answer (*SelfRC*) | 35.86% |
| % of QAs with no noun phrase common between Question and Answer (*ParaphraseRC*) | 49.76% |
| Avg. Minimum Distance between Named Entities in Question and Answer (in the *Self* and *ParaphraseRC* Plot) | Avg distance 16 words more for *ParaphraseRC* |
| % Length of the Longest Common Subsequence of Non-stop words in Query and *SelfRC* plot | 38.1% of the query |
| % Length of the Longest Common Subsequence of Non-stop words in Query and *ParaphraseRC* plot | 21.9% of the query |

**Table 1:** Statistics regarding the *Self* and *ParaphraseRC* Datasets

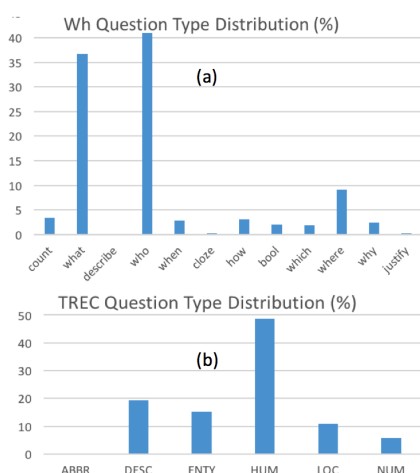

**Figure 2:** Analysis of the Question Types

We refer to this combined dataset containing a total of 186,089 instances as *DuoRC*. Fig. 2 shows the distribution of different Wh-type questions in our dataset. Some more interesting statistics about the dataset are presented in Table 1 and also in Appendix B.

Another notable observation is that in many cases the answers to the same question are different in the two versions. Specifically, only 40.7% of the questions have the same answer in the two documents. For around 37.8% of the questions there is no overlap between the words in the two answers. For the remaining 21% of the questions there is a partial overlap between the two answers. For e.g., the answer derived from the shorter version could be "using his wife's gun" and from the longer version could be "with Dana's handgun" where Dana is the name of the wife. In Appendix A, we provide a few randomly picked examples from our dataset which should convince the reader of the difficulty of *ParaphraseRC* and its differences with *SelfRC*.

# 4 MODELS

In this section, we describe in detail the various state-of-the-art RC and language generation models along with a collection of traditional NLP techniques employed together that will serve to establish baseline performance on the DuoRC dataset.

Most of the current state-of-the-art models for RC assume that the answer corresponds to a span in the document and the task of the model is to predict this span. This is indeed true for the SQuAD, TriviaQA and NewsQA datasets. However, in our dataset, in many cases the answers do not correspond to an exact span in the document but are synthesized by humans. Specifically, for the *SelfRC* version of the dataset around 30% of the answers are synthesized and do not match a span in the document whereas for the *ParaphraseRC* task this number is 50%. Nevertheless, we could still leverage the advances made on the SQuAD dataset and adapt these span prediction models for our task. To do so, we propose to use two models. The first model is a basic span prediction model which we train and evaluate using only those instances in our dataset where the answer matches a span in the document. The purpose of this model is to establish whether even for instances where the answer matches a span in the document, our dataset is harder than the SQuAD dataset or not. Specifically, we want to explore the performance of state-of-the-art models (such as DCN (Xiong et al., 2016)), which exhibit near human results on the SQuAD dataset, on DuoRC (especially, in the *ParaphraseRC* setup). To do so, we seek to employ a good span prediction model for which (i) the performance is within 3-5% of the top performing model on the SQuAD leaderboard (Rajpurkar et al., 2016b) and (ii) the results are reproducible based on the code released by the authors of the paper. Note that the second criteria is important to ensure that the poor performance of the model is not due to incorrect implementation. The Bidirectional Attention Flow (BiDAF) model (Seo et al., 2016) satisfies these criteria and hence we employ this model. Due to space constraints, we do not provide details of the

BiDAF model here and simply refer the reader to the original paper. In the remainder of this paper we will refer to this model as the *SpanModel*.

The second model that we employ is a two stage process which first predicts the span and then synthesizes the answers from the span. Here again, for the first step (*i.e.*, span prediction) we use the BiDAF model (Seo et al., 2016). The job of the second model is to then take the span (mini-document) and question (query) as input and generate the answer. For this, we employ a state-of-the-art query based abstractive summarization model (Nema et al., 2017) as this task is very similar to our task. Specifically, in query based abstractive summarization the training data is of the form {query, document, generated_summary} and in our case the training data is of the form {query, mini-document, generated_answer}. Once again we refer the reader to the original paper (Nema et al., 2017) for details of the model. We refer to this two stage model as the *GenModel*.

Note that Tan et al. (2017) recently proposed an answer generation model for the MS MARCO dataset. However, the authors have not released their code and therefore, in the interest of reproducibility of our work, we omit incorporating this model in this paper.

**Additional NLP pre-processing:**   Referring back to the example cited in Fig. 1, we reiterate that ideally a good model for *ParaphraseRC* would require: (i) employing a knowledge graph, (ii) common-sense knowledge (iii) paraphrase/semantic understanding (iv) multiple-sentence inferencing across events in the passage including coreference resolution of named entities and nouns, and (v) educated guesswork when the question is not directly answerable but there are subtle hints in the passage. While addressing all of these challenges in their entirety is beyond the scope of a single paper, in the interest of establishing a good baseline for DuoRC, we additionally seek to address some of these challenges to a certain extent by using standard NLP techniques. Specifically, we look at the problems of paraphrase understanding, coreference resolution and handling long passages.

To do so, we prune the document and extract only those sentences which are most relevant to the question, so that the span detector does not need to look at the entire 900-word long *ParaphraseRC* plot. Now, since these relevant sentences are obtained not from the original but the paraphrased version of the document, they may have a very small word overlap with the question. For example, the question might contain the word "hand gun" and the relevant sentence in the document may contain the word "revolver". Further some of the named entities in the question may not be exactly present in the relevant sentence but may simply be co-referenced. To resolve these coreferences, we first employ the Stanford coreference resolution on the entire document. We then compute the fraction of words in a sentence which match a query word (ignoring stop words). Two words are considered to match if (a) they have the same surface form, or (b) one words is an inflected form of the word (e.g., river and rivers), or (c) the Glove and Skip-thought embeddings of the two words are very close to each other, or (d) the two words appear in the same synset in Wordnet. We consider a sentence to be relevant for the question if at least 50% of the query words (ignoring stop words) match the words in the sentence. If none of the sentences in the document have atleast 50% overlap with the question, then we pick sentences having atleast a 30% overlap with the question.

## 5   EXPERIMENTAL SETUP

In the following sub-sections we describe (i) the evaluation metrics, and (ii) the choices considered for augmenting the training data for the answer generation model. Note that when creating the train, validation and test set, we ensure that the test set does not contain question-answer pairs for any movie that was seen during training. We split the movies in such a way that the resulting train, valid, test sets respectively contain 70%, 15% and 15% of the total number of QA pairs.

**Span-Based Test Set and Full Test Set**   As mentioned earlier, the *SpanModel* only predicts the span in the document whereas the *GenModel* generates the answer after predicting the span. Ideally, the *SpanModel* should only be evaluated on those instances in the test set where the answer matches a span in the document. We refer to this subset of the test set as the *Span-based Test Set*. Though not ideal, we also evaluate the *SpanModel* model on the entire test set. We say this is not ideal because we know for sure that there are many answers in the test set which do not correspond to a span in the document whereas the model was only trained to predict spans. We refer to this as the *Full Test Set*. We also evaluate the *GenModel* on both the test sets.

**Training Data for the GenModel**    As mentioned earlier, the *GenModel* contains two stages; the first stage predicts the span and the second stage then generates an answer from the predicted span. For the first step we plug-in the best performing *SpanModel* from our earlier exploration. To train the second stage we need training data of the form $\{x = span, y= answer\}$ which comes from two types of instances: one where the answer matches a span and the other where the answer is synthesized and the span corresponding to it is not known. In the first case $x=y$ and there is nothing interesting for the model to learn (except for copying the input to the output). In the second case $x$ is not known. To overcome this problem, for the second type of instances, we consider various approaches for finding the approximate span from which the answer could have been generated, in order to augment the training data with $\{x = approx\_span, y= answer\}$ pairs.

The easiest method was to simply treat the entire document as the true span from which the answer was generated ($x = document, y = answer$). The second alternative that we tried was to first extract the named entities, noun phrases and verb phrases from the question and create a lucene query from these components. We then used the lucene search engine to extract the most relevant portions of the document given this query. We then considered this portion of the document as the true span (as opposed to treating the entire document as the true span). Note that lucene could return multiple relevant spans in which case we treat all these $\{x = approx\_span, y= answer\}$ as training instances. Another alternative was to find the longest common subsequence (LCS) between the document and the question and treat this subsequence as the span from which the answer was generated. Of these, we found that the model trained using $\{x = approx\_span, y= answer\}$ pairs created using the LCS based method gave the best results. We report numbers only for this model.

**Evaluation Metrics**    Similar to Rajpurkar et al. (2016a) we use Accuracy and F-score as the evaluation metric. While accuracy, being a stricter metric, considers a predicted answer to be correct only if it exactly matches the true answer, F-score also gives credit to predictions partially overlapping with the true answer.

## 6    RESULTS AND DISCUSSIONS

The results of our experiments are summarized in Tables 2 to 4 which we discuss in the following sub-sections.

| Preprocessing step of Relevant Sub-plot Extraction | Plot Compression | Answer Recall |
|---|---|---|
| Using WordNet synonym + Glove based paraphrase | 30% | 66.51% |
| WordNet synonym + Glove based paraphrase on Coref resolved plots | 50% | 84.10% |
| WordNet synonym + Glove + Skip-thought based paraphrase on Coref resolved plots | 48% | 85% |

**Table 2:** Performance of the preprocessing step. Plot compression is the % size of the extracted plot w.r.t the original plot size

| SelfRC | Span Test subset | | Full Test set | |
|---|---|---|---|---|
| | Accur. | F1 | Accur. | F1 |
| SpanModel | 46.14 | 57.49 | 37.53 | 50.56 |
| GenModel (with augmented training data) | 16.45 | 26.97 | 15.31 | 24.05 |

| ParaphraseRC | Span Test subset | | Full Test set | |
|---|---|---|---|---|
| | Accur. | F1 | Accur. | F1 |
| SpanModel | 17.93 | 26.27 | 9.78 | 16.33 |
| SpanModel with Preprocessed Data | 27.49 | 35.10 | 14.92 | 21.53 |
| GenModel (with augmented training data) | 12.66 | 19.48 | 5.42 | 9.64 |

**Table 3:** Performance of the *SpanModel* and *GenModel* on the Span Test subset and the Full Test Set of the *Self* and *ParaphraseRC*.

| Train On | Test On | Span Test Set | | Full Test Set | |
|---|---|---|---|---|---|
| | | Accur. | F1 | Accur. | F1 |
| Self RC | Self RC | 46.14 | 57.49 | 37.53 | 50.56 |
| | Paraphrase RC | 27.85 | 36.82 | 15.16 | 22.70 |
| | Self RC + Paraphrase RC | 37.79 | 48.05 | 25.05 | 35.01 |
| Paraphrase RC | Self RC | 34.85 | 45.71 | 28.25 | 40.16 |
| | Paraphrase RC | 19.74 | 27.57 | 10.78 | 17.13 |
| | Self RC + Paraphrase RC | 27.94 | 37.42 | 18.50 | 27.31 |
| Self RC + Paraphrase RC | Self RC | 49.66 | 61.45 | 40.24 | 54.04 |
| | Paraphrase RC | 29.88 | 39.34 | 16.33 | 24.25 |
| | Self + Paraphrase RC | 40.62 | 51.35 | 26.90 | 37.42 |

**Table 4:** Combined and Cross-Testing between *Self* and *ParaphraseRC* Dataset, by taking the best performing *SpanModel* from Table 3.

- *SpanModel* **v/s** *GenModel*: Comparing the first two rows (*SelfRC*) and the last two rows (*ParaphraseRC*) of Table 3 we see that the *SpanModel* clearly outperforms the *GenModel*. This is not very surprising for two reasons. First, around 70% (and 50%) of the answers in *SelfRC* (and *ParaphraseRC*) respectively, match an exact span in the document so the span based model still has scope to do well on these answers. On the other hand, even if the first stage of the *GenModel* predicts the span correctly, the second stage could make an error in generating the correct answer from it because generation is a harder problem. For the second stage, it is expected that the *GenModel* should learn to copy the predicted span to produce the answer output (as is required in most cases) and only occasionally where necessary, generate an answer. However, surprisingly the *GenModel* fails to even do this. Manual inspection of the generated answers shows that in many cases the generator ends up generating either more or fewer words compared the true answer. This demonstrates that there is clearly scope for the *GenModel* to perform better.

- **SelfRC v/s ParaphraseRC:** Comparing the *SelfRC* and *ParaphraseRC* numbers in Table 3, we observe that the performance of the models clearly drops for the latter task, thus validating our hypothesis that *ParaphraseRC* is a indeed a much harder task.

- **Effect of NLP pre-processing:** As mentioned in Section 4, for *ParaphraseRC*, we first perform a few pre-processing steps to identify relevant sentences in the longer document. In order to evaluate whether the pre-processing method is effective, we compute: (i) the percentage of the document that gets pruned, and (ii) whether the true answer is present in the pruned document (i.e., average recall of the answer). We can compute the recall only for the span-based subset of the data since for the remaining data we do not know the true span. In Table 2, we report these two quantities for the span-based subset using different pruning strategies. Finally, comparing the *SpanModel* with and without Paraphrasing in Table 3 for *ParaphraseRC*, we observe that the pre-processing step indeed improves the performance of the Span Detection Model.

- **Effect of oracle pre-processing:** As noted in Section 3, the *ParaphraseRC* plot is almost double in length in comparison to the *SelfRC* plot, which while adding to the complexities of the former task, is clearly not the primary reason of the model's poor performance on that. To empirically validate this, we perform an Oracle pre-processing step, where, starting with the knowledge of the span containing the true answer, we extract a subplot around it such that the span is randomly located within that subplot and the average length of the subplot is similar to the *SelfRC* plots. The *SpanModel* with this Oracle preprocessed data exhibits a minor improvement in performance over that with rule-based preprocessing (1.6% in Accuracy and 4.3% in F1 over the Span Test), still failing to bridge the wide performance gap between the *SelfRC* and *ParaphraseRC* task.

- **Cross Testing** We wanted to examine whether a model trained on *SelfRC* performs well on *ParaphraseRC* and vice-versa. We also wanted to evaluate if merging the two datasets improves the performance of the model. For this we experimented with various combinations of train and test data. The results of these experiments for the *SpanModel* are summarized in Table 4. We make two main observations. First, training on one dataset and evaluating on the other results in a drop in the performance. Merging the training data from the two datasets exhibits better performance on the individual test sets.

Based on our experiments and empirical observations we believe that the *DuoRC* dataset indeed holds a lot of potential for advancing the horizon of complex language understanding by exposing newer challenges in this area.

## 7 CONCLUSION

In this paper we introduced DuoRC, a large scale RC dataset of 186K human-generated question-answer pairs created from 7680 pairs of parallel movie-plots, each pair taken from Wikipedia and IMDb. We then showed that this dataset, by design, ensures very little or no lexical overlap between the questions created from one version and the segments containing the answer in the other version. With this, we hope to introduce the RC community to new research challenges on question-answering requiring external knowledge and common-sense driven reasoning, deeper language understanding and multiple-sentence inferencing. Through our experiments, we show how the state-of-the-art RC models, which have achieved near human performance on the SQuAD dataset, perform poorly on our dataset, thus emphasizing the need to explore further avenues for research.

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

## APPENDIX A  EXAMPLES

In this appendix, we showcase some examples of plots from which questions are created and answered. Since the questions are created from the smaller plot, answering these questions by the reading the smaller plot (which is named as the *SelfRC* task) is straightforward. However, answering them by reading the larger plot (i.e. the *ParaphraseRC* task) is more challenging and requires multi-sentence and sometimes multi-paragraph inferencing.

Due to shortage of space, we truncate the plot contents and only show snippets from which the questions can be answered. In the smaller plot, blue indicates that an answer can directly be found from the sentence and cyan indicates that the answer spans over multiple sentences. For the larger plot, red and orange are used respectively.

### A.1  EXAMPLE 1: PALE RIDER (1985)

#### A.1.1  SMALLER PLOT

In the countryside outside the fictional town of Lahood, California, sometime around 1880, [[thugs working for big-time miner Coy LaHood ride in and destroy the camp of a group of struggling miners]$^{Q1}$ and their families who have settled in nearby Carbon Canyon and are panning for gold there. In leaving, they also shoot the little dog of fourteen-year-old Megan Wheeler]$^{Q15}$. As Megan buries her dog in the woods and prays for a miracle, a stranger passes by heading to the town on horseback.

[Megan's mother, Sarah]$^{Q16}$, is being courted by [Hull Barret, the leader of the miners]$^{Q17}$ ...[Coy LaHood's son Josh]$^{Q8}$ ...[Club, who with one hammer blow smashes a large rock]$^{Q7}$ ...[Coy LaHood has been away in Sacramento]$^{Q9}$ ...[[Megan, who has grown fond of the Preacher, goes looking for him, but Josh confronts and attempts to rape her]$^{Q11}$, while his cohorts look on and encourage him, except for Club, who sees what is happening and moves forward to help Megan]$^{Q13}$ before Josh can do anything serious. At this moment the [Preacher arrives on horseback armed with a Remington Model 1858 revolver he has recovered from a Wells Fargo office and, after shooting Josh]$^{Q14}$ ...[Stockburn, who appears startled and says that he sounds like someone that he once knew, but that couldn't be, since that man is dead]$^{Q5}$. [Stockburn and his men gun down Spider Conway]$^{Q4}$, ...

[The Preacher and Hull go to LaHood's strip mining site and blow it up with dynamite]$^{Q6}$. [To stop Hull from following him, the Preacher then scares off Hull's horse]$^{Q3}$ and rides into town alone...[Coy LaHood, watching from his office]$^{Q10}$, ...[snow-covered mountains]$^{Q2}$. [Megan then drives into town and shouts her love to the Preacher]$^{Q12}$ and thanks after him. The words echo along the ravine that he is traversing.

#### A.1.2  LARGER PLOT

Somewhere in California, at the end of the Gold Rush, several horsemen come riding down from the nearby mountains ...[The horsemen shoot cattle and Megan's dog]$^{Q4,Q15}$, and then chase donkeys as they leave ...

Hull describes the fight between the stranger and McGill and his men. [Megan's mother, Sarah]$^{Q16}$ says he sounds no different from McGill, Tyson, or any of LaHood's roughnecks ...Preacher says there is lot of sinners around, that he can't leave before he finishes his work. [Josh says, "Club", who gets down and walks into the stream. Everyone is apprehensive. He rolls down his sleeves, and then...quickly grabs Hull's sledgehammer with one hand and strikes the boulder once, screaming, splitting it]$^{Q7}$ ...

[A train pulls into the station from Sacramento while Josh and McGill wait. [Josh's father Coy LaHood]]$^{Q8,Q9}$ (Richard Dysart) exits the train, and then he goes with Josh and McGill...

Josh asks what she really came for. She replies that she's just riding, taking a look around. [Josh says he wants to take a look too, at her real close. He pulls her off the horse. She screams as he carries her downhill...Josh grabs her hair and kisses her. They both fall to the ground. The men cheer him on while Megan begs him to stop]$^{Q11}$ ...a gunshot sounds out. Josh gets up and everyone turns around. [Preacher, on his horse...His gun is trained on Josh. Megan sees him and smiles]$^{Q13}$, ...[Josh falls

| | Question | Shorter Plot Answer | Larger Plot Answer |
|---|---|---|---|
| Q1 | For which big-time miner are the thugs who destroyed miners camp in Carbon Canyon working for? | Coy Lahood | Coy Lahood |
| Q2 | How are the mountains in the film? | Covered with snow | snow-capped |
| Q3 | How does the Preacher stop Hull from following him? | Scares Hulls' horse | To stop Hull from following him, the Preacher then scares off Hull's horse and rides into town alone |
| Q4 | In the movie, who do Stockburn and his men gun down? | Spider Conway | Megan's dog and cattle |
| Q5 | In the movie, why does Stockburn say that the Preacher could not be the man he once knew? | that man is dead | The man Stockburn once knew is dead |
| Q6 | What did they use to blow up the strip mining site? | Dynamite | dynamite |
| Q7 | What does Club smash? | A rock | A Boulder |
| Q8 | What is Coy Lahood's relation to Josh? | Father and son | Father |
| Q9 | Where has Coy Lahood been living? | Sacramento | Sacramento |
| Q10 | Where was Coy watching from? | Office | a window |
| Q11 | Who attempts to rape Megan? | Josh | Josh |
| Q12 | Who does megan love? | The preacher | The preacher |
| Q13 | Who prevents Josh from raping Megan? | Club | the preacher |
| Q14 | Who shoots Josh in the hand? | Preacher | The Preacher |
| Q15 | Whose little dog did the thugs shoot? | Megan Wheeler | Megan |
| Q16 | Who is Megan's mother? | Sarah | Sarah |
| Q17 | Who is the leader of the miners? | Hull Barret | Coy LaHood |

**Table 5:** QA for Pale Rider

to the ground. He reaches for his gun, but Preacher shoots his hand]Q14 . . . LaHood replies, "Tall. Lean. His eyes. . . his eyes. Something strange about em. That mean something to you?" [Stockburn says that it sounds like a man he knew, but that man is dead]Q5 . . . [LaHood watches through the window]Q10 as they kill Spider, . . .

Hull insists on going with him so Preacher agrees. [They go to the LaHood camp and blow up their pipes, sluices, tents, and the barracks with dynamite]Q6. [After fooling Hull to dismount, Preacher scares away his horse. He then tells Hull to take care of Sarah and Megan, and rides into town]Q3 . . . Blankenship tells her that the horses are exhausted and she would kill them. [Megan runs to the end of town and shouts out thank you to Preacher, that they love him, that she loves him]Q12 . . . The final shot of the movie shows Preacher riding through the [snow in the mountains]Q2.

## A.2  EXAMPLE 2: BIG JAKE (1971)

### A.2.1  SMALLER PLOT

[In 1909]$^{Q6}$, [there is a raid on the McCandles family ... Martha, the head of the family ...[In consequence, she sends for her estranged husband, the aging Jacob "Big Jake" McCandles]$^{Q9}$, ...[the ransom to the kidnappers, a million dollars]$^{Q4}$

...[The Texas Ranger captain is present and offers the services of his men]$^{Q11}$, ...Jake, preferring the old ways, has followed on horseback, accompanied by an old Apache associate, Sam Sharpnose. [He is now joined by his sons, Michael and James]$^{Q2}$, ...Knowing that they have been followed by another gang intent on stealing the strongbox, [Jake sets a trap for them and they are all killed]$^{Q16}$. [During the attack, the chest is blasted open]$^{Q1}$, [revealing clipped bundles of newspaper instead of money]$^{Q5}$ ...

A thunderstorm breaks and [Pop Dawson, one of the outlaws, arrives to give them the details of the exchange]$^{Q7}$ ...[Jake arranges for Michael to follow after them to take care of the sharp] ...[Jake tosses the key of the chest to Fain, who opens it up to discover that he has been tricked]$^{Q12}$. [Fain orders his brother Will to kill the boy]$^{Q13}$ but he is shot by Jake. [Dog is wounded by the sniper]$^{Q15}$ and Jake is wounded in the leg before Michael kills him. Jake tells the boy to escape but Little Jake is hunted by the machete wielding [John Goodfellow, who has already hacked Sam to death]$^{Q10}$ ...[With Little Jake rescued, and the broken family bonded, they prepare to head home]$^{Q3}$.

### A.2.2  LARGER PLOT

[[Jacob McCandles (John Wayne) is a big man with a bigger reputation. A successful rancher and landowner]$^{Q14}$, his many businesses keep him conveniently away from his home and estranged wife Martha (Maureen O'Hara)]$^{Q9}$..., [and is demanding one million dollars for his safe return]$^{Q4}$...

[The local sheriff has convened a posse complete with then state-of-the-art automobiles. [Two of Jake's sons, the passionate, gunslinging James (Patrick Wayne) and the motorcycle-riding, sharp-shooting Michael (Christopher Mitchum)]$^{Q2}$ elect to go with the sheriff's posse. Big Jake decides to set off across the rough terrain on his horse with his Dog at his side, and soon meets up with his Native American friend, Sam Sharpnose (Bruce Cabot), who has brought additional horses and supplies]$^{Q11}$.

[They then devise their strategy: James will go have a good time in the saloon, Big Jake will head to the barbershop for a shower, Sam will secrete himself on the roof of the hotel, seemingly leaving Michael alone protecting the strong box. Big Jake tells Sam to listen for a "disturbance" in the street, and use the distraction to join Michael in the hotel room to protect the strong box. As Big Jake predicted, the gang tries to hit the strong box when it looks most vulnerable. Fain and another of his gang members start a fight with James in the saloon, one keeps a gun on Big Jake in the barbershop, and two others come up the hotel stairs and toward the room. Big Jake dispatches his captor in the barbershop, James fights his way out of the saloon with Jake's help, and the two head to the hotel. At the hotel, once Sam hears the fight in the saloon, he climbs over the roof and slips in the window to aid Michael in protecting the strong box. Shotguns blast as the gang hits the hotel room. When James and Jake arrive they find Sam, Michael and the Dog unharmed, [but the strong box has suffered damage. To their horror, James and Michael realize they've been risking their lives to protect a box of newspaper clippings!]]$^{Q1,Q5,Q16}$ ...

[Big Jake takes the few moments he has to plan with his sons. He tells Michael of the sharpshooter and instructs him to find a high position and take him out whenever he can. Big Jake takes the Dog and goes to the meet as instructed, while the others follow discreetly behind]$^{Q8}$ ...[As Fain unlocks the strong box, he realizes he´s been had]$^{Q12}$. Big Jake whispers to him that no matter what happens, Fain will be the first one to die. [Fain screams his command to kill the boy]$^{Q13}$, ...

[Dog giving his life protecting Little Jake from one of Fain's machete-wielding gang]$^{Q15}$...Sam points him toward James at the exit, [who helps Little Jake escape. Fain and Big Jake are in a duel to the death, when Michael takes a fatal shot at Fain, saving his father and Little Jake. After a harrowing journey and a risky gamble, the family leaves, happy to be together]$^{Q3}$.

| | Question | Shorter Plot Answer | Larger Plot Answer |
|---|---|---|---|
| Q1 | How was the strongbox opened? | It was blasted during an attack | it was damaged during the fight |
| Q2 | What are the names of Jake's sons? | Jake's sons are Michael and James | James and Michael |
| Q3 | What did the family prepare to do once Jake had been rescued? | head home | Leave |
| Q4 | What is the amount of the ransom? | The ransom amount is a million dollars | one million dollars |
| Q5 | What was in the strongbox? | Clipped bundles of newspaper | newspaper clippings |
| Q6 | What year does the movie take place? | 1909 | **No Answer** |
| Q7 | Which outlaw gives details of the exchange to the others? | Pop Dawson | **No Answer** |
| Q8 | Who does Jake arrange to follow the rest of the group? | Michael | Michael |
| Q9 | Who is married to Big Jake? | Martha | Martha |
| Q10 | Who killed Sam? | John Goodfellow | **No Answer** |
| Q11 | Who offers his services to help Jake combat the kidnappers? | The Texas ranger captain offers the services of men | the posse, native American friend and his two sons |
| Q12 | Who opens the chest to discover he has been tricked? | Fain | Fain |
| Q13 | Who orders Will to kill the boy? | Fain | Fain |
| Q14 | Who owns the ranch? | McCandles family | McCandles family |
| Q15 | Who wounded the dog? | A sniper | Fain's machete-wielding gang |
| Q16 | Why does Jake set a trap and kill another gang? | They were intent on stealing the strongbox | to protect the strongbox |

**Table 6:** QA for Big Jake

## APPENDIX B  DATA ANALYSIS

We conducted a manual verification of 100 question-answer pairs where the *SelfRC* and *ParaphraseRC* were different or the latter was marked as non-answerable. As noted in Fig. 3, the chief reason behind getting *No Answer* from the Paraphrase plot is lack of information and at times, need for an educated guesswork or missing general knowledge (e.g. Philadelphia is a city) or missing movie meta-data (e.g. to answer questions like 'Where did Julia Roberts' character work in the movie?'). On the other hand, *SelfRC* and *ParaphraseRC* answers are occasionally seen to have partial or no overlap, mainly because of the following causes; phrasal paraphrases or subjective questions (e.g. Why and How type questions) or different valid answers to objective questions (e.g. 'Where did Jane work?' is answered by one worker as 'Bloomberg' and other as 'New York City') or differently spelt names in the answers (e.g. 'Rebeca' as opposed to 'Rebecca').

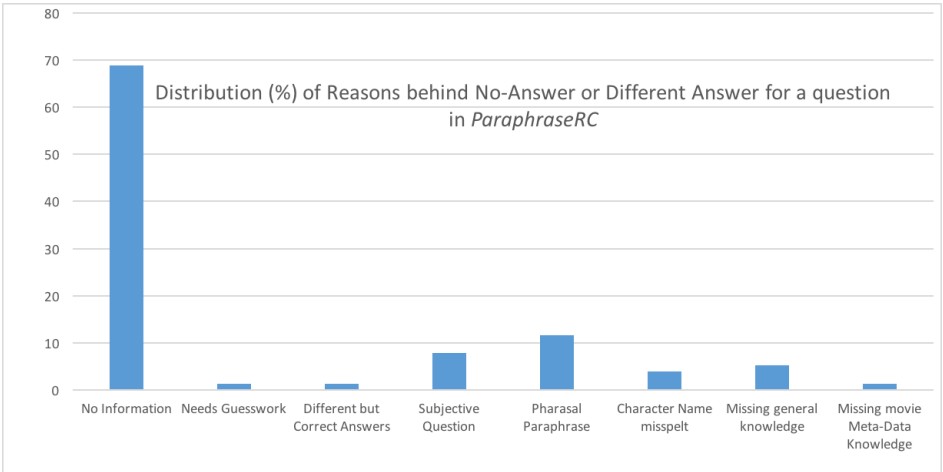

**Figure 3:** Manual Analysis of 100 Questions and their corresponding answers from the *SelfRC* and *ParaphraseRC* Dataset to understand the various reasons behind these two answers being different or the latter being non-answerable

## APPENDIX C  MODEL ARCHITECTURE

Fig. 4 illustrates the 5-step process of answering a question from the comprehension, by optionally pre-processing the input passage in Step 2 and 3, then using the BiDirectional Attention Flow (BiDAF) model for Span identification, and finally generating the answer text from the identified span by employing a state-of-the-art query-based Abstractive Summarization (qBAS) model.

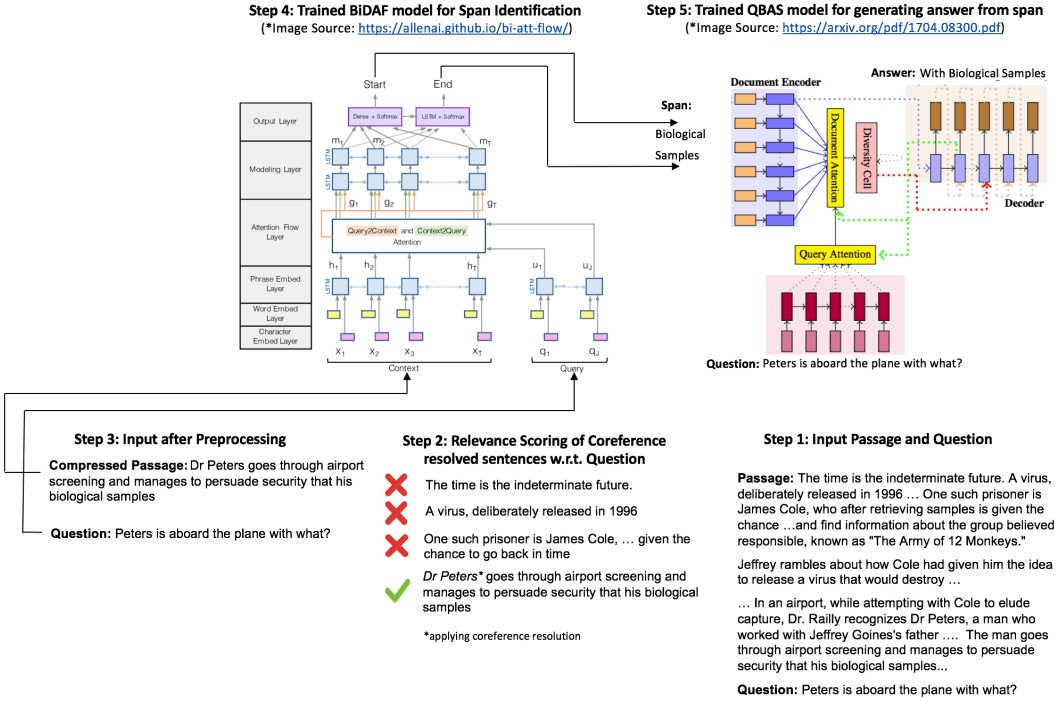

**Figure 4:** Model architecture

## APPENDIX D    PERFORMANCE ANALYSIS

In Fig. 5 we show a performance analysis of the *SelfRC* and *ParaphraseRC* tasks when evaluated on the Span Test Subset and the Full Test Set, over different question types and plots of different length.

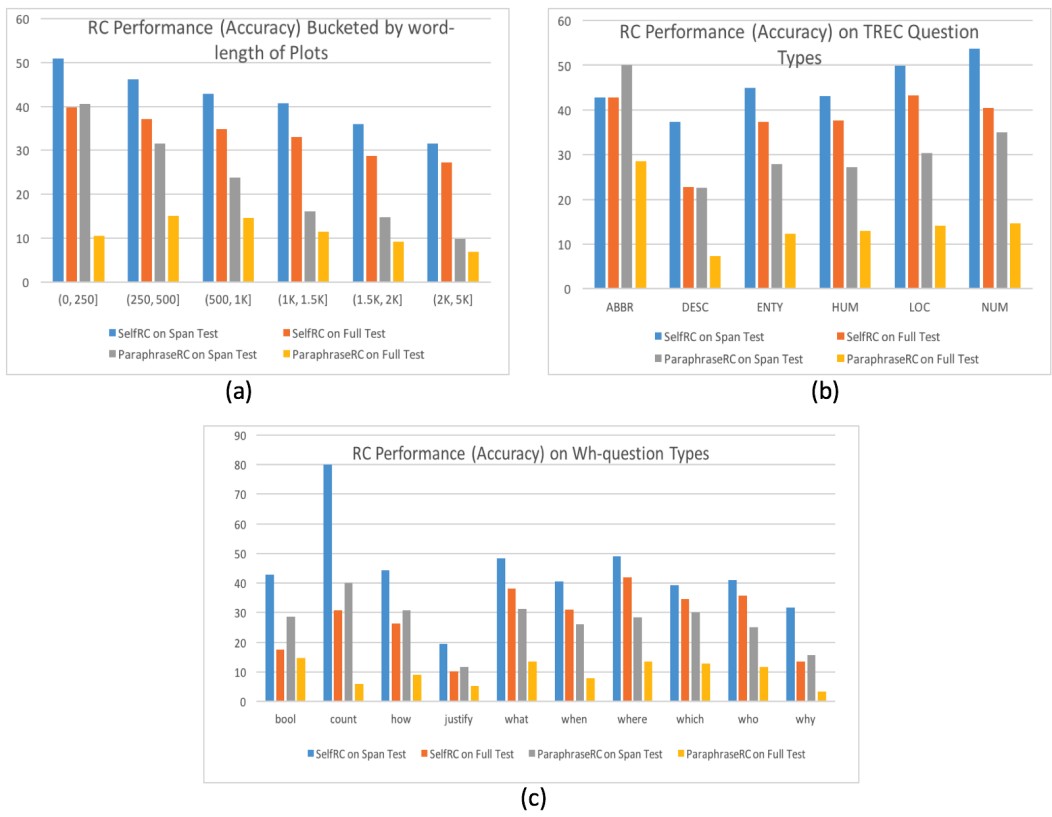

**Figure 5:** Performance Analysis of the *Self* and *ParaphraseRC* on different plot-lengths or different question-types

