# OpenReview forum: "DuoRC: Towards Complex Language Understanding with Paraphrased Reading Comprehension"
_ICLR.cc/2018/Conference — Invite to Workshop Track_

### Official Review · AnonReviewer3 · 2017-11-27
**Useful dataset for reading comprehension**

**Rating:** 7
**Confidence:** 4

**Review:**

This paper presents a useful dataset for testing reading comprehension while avoiding significant lexical overlap between question and document. The paper rightly mentions that existing reading comprehension datasets (e.g. SQuAD) where the current methods are already performing at the human level largely due to large lexical overlap between question and document. The authors have devised a clever way to create a reading comprehension dataset without a lot of lexical overlap by using parallel plots of movies from Wikipedia and IMDB.

This paper contributes a useful new dataset that fixes some of the shortcomings of existing reading comprehension datasets where the task is made easier by lexical overlap. The authors also present an analysis of the data by applying one of the SOTA techniques on SQuAD to this data. They also analyze the effect of various span-identification steps and preprocessing steps on the performance. Overall, this paper contributes a useful new dataset that can be quite useful for reading comprehension.

---

> ### Author Response · Authors · 2017-12-11
> **Addressing AnonReviewer3's comments**
>
> Thank you for the encouraging words and appreciating the usefulness of the dataset

---

### Official Review · AnonReviewer2 · 2017-11-27
**Useful dataset for reading comprehension**

**Rating:** 6
**Confidence:** 4

**Review:**

1) This paper proposes a new dataset for Reading Comprehension (RC). Different from other existing RC datasets, the authors claim that this new dataset requires background and common-sense knowledge,  and across sentences reasoning in order to answer the questions correctly.

Overall, I think this dataset is very useful for RC. The collection process is also carefully designed to reduce the lexical overlap between question and answer pairs.

2) I have the questions as follows:
i) in the abstract, authors mentioned the workers set one only takes care of creating questions from version one of the plots, and workers set two is in charge of generating answers from another version of plots. However, in bullet 2 of section 3, it seems that the workers set one is also required to answer the questions in selfRC. Is there any mistake in the description of the abstract?

ii) What is the standard for creating the questions? I noticed that the time and location information was used to generate questions sometime, but sometimes these kinds of questions are ignored.

iii) Why the SelfRC is about QA pairs but for ParaphraseRC, you need to include documents?

iv) What is the average length of the answers in both ParaphraseRC and SelfRC? I found that the answers are usually very short, which is more like factoid QA. It would be great if the authors could design some non-factoid QA pairs which require more reasoning and background knowledge.

v) During NLP pre-processing (section 4), how do you prune the irrelevant documents?

---

> ### Author Response · Authors · 2017-12-10
> **Addressing AnonReviewer2's comments**
>
> We would like to thank the reviewer for the valuable comments and would take this opportunity to address the specific comments/questions raised.
>
> 1. in the abstract, authors mentioned the workers set one only takes care of creating questions from version one of the plots, and workers set two is in charge of generating answers from another version of plots. However, in bullet 2 of section 3, it seems that the workers set one is also required to answer the questions in selfRC. Is there any mistake in the description of the abstract?
>
> RESPONSE: We thank the reviewer for pointing this out. The worker set one are responsible for creating QA pairs from the selfRC plot and the set two workers are in charge of generating the answers from the ParaphraseRC plot. We will correct the statement in the abstract.
>
>
> 2. Why the SelfRC is about QA pairs but for ParaphraseRC, you need to include documents?
>
> RESPONSE:  Let A and B be two versions of the same movie plot. selfRC is about creating both questions and answers from plot A. ParaphraseRC is about reading plot B and trying to answer the question created using plot A. Therefore, both selfRC and paraphraseRC requires a document(plot) to answer questions.
>
> 3. What is the average length of the answers in both ParaphraseRC and SelfRC? I found that the answers are usually very short, which is more like factoid QA. It would be great if the authors could design some non-factoid QA pairs which require more reasoning and background knowledge.
>
> RESPONSE: The average answer length for SelfRC is 3 words and for ParaphraseRC is 5 words. Apart from the factual questions we also have 7% how/why/justify/describe type questions, 6% boolean/count questions and 1% cloze questions. Even though the majority of the questions are factoid (what/who/when/which/where) the complexity of the ParaphraseRC dataset arises from the fact that for 37% of the questions the answer is not directly present in the plot and has to “synthesized” based on the plot information and an additional 13% of the questions are entirely non-answerable. Another important aspect of the complexity is due to the poor textual overlap between the plot and the words in the QA. In ParaphraseRC, only 12% of the QA pairs have non-zero textual overlap between the plot and both the question-words and answer words.
>
> 4. During NLP pre-processing (section 4), how do you prune the irrelevant documents?
>
> RESPONSE: We do not prune irrelevant documents but irrelevant segments (sentences or paragraphs) from the given plot based on semantic relation between the words in the plot segments and the question. This preprocessing step is elaborated in the Subsection titled “Additional NLP pre-processing” in section 4.

---

### Official Review · AnonReviewer1 · 2017-11-30
**Need some more analysis / clarifications.**

**Rating:** 7
**Confidence:** 3

**Review:**

Summary:
The paper proposes a new dataset for reading comprehension, called DuoRC. The questions and answers in the DuoRC dataset are created from different versions of a movie plot narrating the same underlying story. The DuoRC dataset offers the following challenges compared to the existing reading comprehension (RC) datasets – 1) low lexical overlap between questions and their corresponding passages, 2) requires use of common-sense knowledge to answer the question, 3) requires reasoning across multiples sentences to answer the question, 4) consists of those questions as well that cannot be answered from the given passage. The paper experiments with two types of models – 1) a model which only predicts the span in a document and 2) a model which generates the answer after predicting the span. Both these models are built off of an existing model on SQuAD – the Bidirectional Attention Flow (BiDAF) model. The experimental results show that the span based model performs better than the model which generates the answers. But the accuracy of both the models is significantly lower than that of their base model (BiDAF) on SQuAD, demonstrating the difficulty of the DuoRC dataset.

Strengths:

1.	The data collection process is interesting. The challenges in the proposed dataset as outlined in the paper seem worth pushing for.
2.	The paper is well written making it easy to follow.
3.	The experiments and analysis presented in the paper are insightful.

Weaknesses:

1.	It would be good if the paper can throw some more light on the comparison between the existing MovieQA dataset and the proposed DuoRC dataset, other than the size.
2.	The dataset is motivated as consisting of four challenges (described in the summary above) that do not exist in the existing RC datasets. However, the paper lacks an analysis on what percentage of questions in the proposed dataset belong to each category of the four challenges. Such an analysis would helpful to accurately get an estimate of the proportion of these challenges in the dataset.
3.	It is not clear from the paper how should the questions which are unanswerable be evaluated. As in, what should be the ground-truth answer against which the answers should such questions be evaluated. Clearly, string matching would not work because a model could say “don’t know” whereas some other model could say “unanswerable”. So, does the training data have a particular string as the ground truth answer for such questions, so that a model can just be trained to spit out that particular string when it thinks it can’t answer the questions?
4.	One of the observations made in the paper is that “training on one dataset and evaluating on the other results in a drop in the performance.” However, in table 4, evaluating on Paraphrase RC is better when trained on Self RC as opposed to when trained on Paraphrase RC. This seems to be in conflict with the observation drawn in the paper. Could authors please clarify this? Also, could authors please throw some light on why this might be happening?
5.	In the third phase of data collection (Paraphrase RC), was waiting for 2-3 weeks the only step taken in order to ensure that the workers for this stage are different from those in stage 2, or was something more sophisticated implemented which did not allow a worker who has worked in stage 2 to be able to participate in stage 3?
6.	Typo: Dataset section, phrases --> phases

Overall: The challenges proposed in the DuoRC dataset are interesting. The paper is well written and the experiments are interesting. However, there are some questions (as mentioned in the Weaknesses section) which need to be clarified before I can recommend acceptance for the paper.

---

> ### Author Response · Authors · 2017-12-10
> **Addressing AnonReviewer1's comments**
>
> We would like to thank the reviewer for the valuable comments and would take this opportunity to address the specific comments/questions raised.
>
>
> 1. Comparison between MovieQA and ParaphraseRC
>
> More details of this comparative analysis is given below in a separate comment titled “COMPARATIVE STATS FOR MOVIEQA AND DUORC”
> i) MovieQA, like the SQUAD RC Dataset, also suffers from a high lexical overlap between QA pairs and the passage. In particular, the percentage of Questions where both question & answer entities were found in the plot is only 12% in ParaphraseRC whereas it is 65-68% in MovieQA (over the Train and Valid Splits). Similarly, the percentage of questions where question entities were found in the plot is only 47% in ParaphraseRC while its 57-60% in MovieQA.
> ii) Scale of the Data: ParaphraseRC is 6.7 times of MovieQA (in terms of QA pairs)
> iii) Multiple Sentence Inferencing: Both ParaphraseRC and MovieQA require inferencing over 2-3 sentences on an average to answer the questions
>
>
> 2. Distribution of Questions exhibiting the challenges of DuoRC (For more details on each of these points please see the separate comment below, titled "COMPARATIVE STATS FOR MOVIEQA AND DUORC")
>
> Challenge 1 - Low lexical overlap between question and plot: For ParaphraseRC, 47% of the questions have some meaningful overlap with the plot (and on an avg. only 21% of the query entities or noun/verb phrases are present in the plot)
>  Challenge 2 - Questions requiring common-sense knowledge: These are possibly the ones which don’t have any direct textual overlap between the question/answer and the plot content. In Paraphrase RC 88% of the questions require external knowledge to bridge the gap.
>  Challenge 3 - Questions requiring multiple sentence inferencing: On an average answering a question from the ParaphraseRC plot requires inferencing over 2-3 sentences (please see the "num_sentences_req_for_inferencing" stats below in the "COMPARATIVE STATS FOR MOVIEQA AND DUORC" section).
>  Challenge 4 - Questions that require answers to be generated and not just extracted from the passage: 37% of the Questions are “synthesized” by AMT workers after reading the ParaphraseRC plot
>  Challenge 5 - Questions that are “Not Answerable”:  13% of questions could not be answered by AMT workers based on that plot
>  Challenge 6 - Non Factoid questions: Apart from the factual questions we also have 7% how/why/justify/describe type non-factoid questions, 6% boolean/count questions and 1% cloze questions.
>
>
> 3. How to evaluate “Non Answerable” Questions
>
> Yes, the correct answer to a question in our dataset is either: i) a text snippet directly taken from the plot, or ii) a text “synthesized” by the annotator based on the plot, or iii) the question is “Not Answerable” from the plot. Therefore, for each question a model can either: a) predict the likely span containing the answer and/or generate the answer from it, or b) make a prediction as “Not Answerable” (for example, “No Span” output from the BiDAF model). We can separately benchmark the accuracy of any model over the subset of questions which are marked as “Not Answerable”.
>
>
> 4. Evaluating on Paraphrase RC is better when trained on Self RC as opposed to when trained on Paraphrase RC.
>
> We thank the reviewer for pointing out the mistake in the Discussion section that “training on one dataset and evaluating on the other results in a drop in the performance.” is indeed not true in the case where the model is trained on SelfRC and evaluated on ParaphraseRC. We believe this is because learning with the  ParaphraseRC is more difficult given the wide range of challenges in this dataset. However, in our setup, instead of replacing the training data, SelfRC, with ParaphraseRC (which drops the test performance on both SelfRC and ParaphraseRC), if we augment the training data SelfRC with ParaphraseRC, the test performance infact improves slightly indicating that ParaphraseRC also helps to an extent. We will correct this observation in the updated version of the paper.
>
>
> 5. In the third phase of data collection (Paraphrase RC), was waiting for 2-3 weeks the only step taken in order to ensure that the workers for this stage are different from those in stage 2, or was something more sophisticated implemented which did not allow a worker who has worked in stage 2 to be able to participate in stage 3?
>
>  No, this was the only step that was taken. But given the scale of movies (~8K movies) over a diverse set of genres, languages, etc and the global AMT worker base, hopefully this step was sufficient to remove any chance of bias.

---

> > ### Author Response · Authors · 2017-12-10
> > **COMPARATIVE STATS FOR MOVIEQA AND DUORC**
> >
> > To compare MovieQA and DuoRC, we extracted the following statistics. First we extracted entities (which includes named entities and noun or verb phrases) in the question and answer. Then we located sentences in the plot containing these entities. Next,  for each question entity located in a sentence, we find the closest sentences containing the answer entities. From this we derive two things
> > ----“avg_distance_in_words” or "avg_distance_in_sentences" below means the average distance (in terms of words/sentences) between the occurrence of the question entities and closest occurrence of the answer entities.
> > ----"num_sentences_req_for_inferencing", i.e. total number of sentences required to cover all the question and answer entities (only considering the closest occurrence of sentence containing answer entities to the sentence containing question entities)
> >
> > For MovieQA Valid:
> > avg_distance_in_words  20.6 words
> > avg_distance_in_sentences  1.69 sentences
> > num_sentences_req_for_inferencing  2.28 sentences
> > % Qs where both question & answer entities were found in the plot: 1287/1958 i.e. 65.7%
> > % Qs where only question entities found in the plot: 1126/1958 i.e. 57.5%  (Percentage length of LCS (Longest Common Subsequence of non-stop words) between query and plot w.r.t question length: 25% of the query)
> >
> > For MovieQA Train:
> > avg_distance_in_words  20.75 words
> > avg_distance_in_sentences  1.66 sentences
> > num_sentences_req_for_inferencing 2.33 sentences
> > % Qs where both question & answer entities were found in the plot: 6737/9848 i.e. 68.4%,
> > % Qs where only question entities found in the plot: 5912/9848 i.e. 60% (Percentage length of LCS (Longest Common Subsequence of non-stop words) between query and plot w.r.t question length: 25% of the query)
> >
> > For ParaphraseRC:
> > avg_distance_in_words  45.3 words
> > avg_distance_in_sentences  2.7 sentences
> > num_sentences_req_for_inferencing 2.47 sentences
> > % Qs where both question & answer entities were found in the plot: 12294/100316 i.e. 12%,
> > % Qs where only question entities found in the plot: 47198/100316 i.e. 47% (Percentage length of LCS (Longest Common Subsequence of non-stop words) between query and plot w.r.t question length: 21% of the query)
> >
> > For SelfRC:
> > avg_distance_in_words  13.4 words
> > avg_distance_in_sentences  1.34 sentences
> > num_sentences_req_for_inferencing 1.51 sentences
> > % Qs where both question & answer entities were found in the plot: 50423/85773 i.e. 58.7%,
> > % Qs where only question entities found in the plot:  54371/85773 i.e. 63.3% (Percentage length of LCS (Longest Common Subsequence of non-stop words) between query and plot w.r.t question length: 38% of the query)

---

> > > ### Comment · AnonReviewer1 · 2018-01-15
> > > **Post-rebuttal evaluation**
> > >
> > > After reading the authors' responses to the concerns raised by me and my fellow reviewers, I would recommend acceptance of this paper because it presents a new dataset which presents challenges worth pushing for.

---

### Decision · Program_Chairs · 2018-01-29
**ICLR 2018 Conference Acceptance Decision**

**Decision:**

Invite to Workshop Track

**Comment:**

This is a (question answering) dataset paper with some baseline models.

The evaluation metric seems far from ideal and not quite ready for prime-time yet. They use F1 and Exact Match - these metrics make sense for extractive question answering systems, they don't make sense IMO for abstractive systems where the answer can be generated by the model (BLEU-type eval metrics seem more appropriate).

I therefore recommend this work for the workshop track.